# Can you tell that I'm confused? An overhearer study for German backchannels by an embodied agent

ISABEL DONYA MEYWIRTH, Computational Linguistics, Department Linguistics, University of Potsdam, Germany

JANA GÖTZE, Computational Linguistics, Department Linguistics, University of Potsdam, Germany

In spoken interaction, humans constantly display and interpret each others' state of understanding. For an embodied agent, displaying its internal state of understanding in an efficient manner can be an important means for making a user-interaction more natural and initiate error recovery as early as possible. We carry out an overhearer study with 62 participants to investigate whether German verbal and non-verbal backchannels by the virtual Furhat embodied agent can be interpreted by an overhearer of a human-robot conversation. We compare three positive, three negative, and one neutral feedback reaction. We find that even though it is difficult to generate certain verbal backchannels, our participants can recognize displays of understanding with an accuracy of up to 0.92. Trying to communicate a lack of understanding is more often misunderstood (accuracy: 0.55), meaning that interaction designers need to carefully craft them in order to be useful for the interaction flow.

CCS Concepts: • **Human-centered computing → HCI design and evaluation methods**; **Usability testing**; **Empirical studies in HCI**.

Additional Key Words and Phrases: backchannels, grounding in conversation, human-robot interaction, social virtual agents, listener responses

**ACM Reference Format:**
Isabel Donya Meywirth and Jana Götze. 2022. Can you tell that I'm confused? An overhearer study for German backchannels by an embodied agent. In *INTERNATIONAL CONFERENCE ON MULTIMODAL INTERACTION (ICMI '22 Companion), November 7–11, 2022, Bengaluru, India.* ACM, New York, NY, USA, 8 pages. https://doi.org/10.1145/3536220.3558804

## 1 INTRODUCTION

In human-human interaction, listeners are not just passive recipients of information; instead they constantly give feedback [5]. A micro-analysis of human conversations found that listener feedback is rarely internally motivated and instead requested by the speaker through gestural behavior, such as head movements [9]. Non-verbal feedback, summarized as "a 'cooperative' way of exchanging information about the successfulness of communication" by Allwood and Cerrato [1], is crucial in well-functioning human conversations.

A similar case can be made in human-robot interaction (HRI), where a robot or virtual agent might analyze human feedback, but also the human needs information on the agent's internal state. In instruction-giving scenarios it is helpful to receive feedback whether the instructions are being understood or even whether the robot or virtual agent is listening. Feedback should be undisruptive and incremental, as it is favored in human-human conversation [15]. This

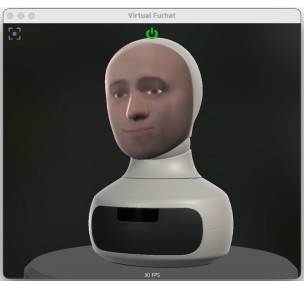

Fig. 1. Screenshot of an overhearer perspective

establishes a need to investigate which verbal and non-verbal feedback strategies, performed by a social embodied virtual agent, are accepted and correctly understood by human adults.

Listener feedback depicts certain grounding acts, e.g., the acceptance of information. Grounding describes the process in which an utterance is put in the Common Ground (CG). The CG is an abstract set of information containing all shared knowledge and beliefs of all involved discourse participants. In human everyday conversation, grounding is needed to carry on a conversation [2]. Implicit information, including information on grounding, is exchanged in the so-called "backchannel" without disrupting the verbal discourse. Yngve [16] introduced the term "backchannel" as a communication channel in the "background".

In this study we investigate whether human participants can recognize verbal and non-verbal backchannels performed by an embodied virtual agent as its intended internal state. Participants in an overhearer role are to distinguish the agent's state in a conversation carried out in German. Head gestures, facial expressions and simple verbal backchannels are enacted by the virtual version of the robotic head Furhat[1] as can be seen in Figure 1.

## 2 RELATED WORK

The perception of different backchannels in human-agent interaction has been the focus of previous studies. In human communication there are several forms of head movements, facial expressions or verbal backchannels and each type can have different meanings depending on the context, manner of expression, personality, etc.

In an online study, Mirnig et al. [10] had participants rate different facial expressions of two humanoid robots, *EDDIE* and *IURO*. In their video stimuli, the robots performed one of the emotions *neutral*, *happy*, *sad* or *surprise* through a facial expression. Their expressions were generated without context, i.e., not reactive to any utterance. The results indicated that the happy condition was better recognized for the *IURO* robot, while the sad and surprised condition showed best results for the *EDDIE* robot. They concluded that the implementation and design of facial expressions on robots is not straightforward and therefore they advise to do prior testing whether the implemented gestures are recognized as intended.

Jonell et al. [7] used a similar evaluation technique to analyze their model for learning non-verbal behavior from YouTube videos on the Furhat robot. For their evaluation they used videos of the physical and the virtual Furhat. In an online study, subjects were requested to watch videos of Furhat performing different gestures learned with Jonell et al. [7]âĂŹs model and rated them on coherence between head movements, voice and appropriateness of the facial expression for a social robot. Their results confirm that the presence of non-verbal behavior for both the physical and

---

[1]https://furhatrobotics.com

Table 1. Distribution of answers and counts (N=122 for all settings, except N=121 for settings 5 and 7). Highest counts are in bold for each setting. We collapse the three positive and negative settings, respectively, to test for differences against random selection with $\chi^2$-tests.

| # | condition | setting | confusion | disinterest | understanding | unattentive | comparison |
|---|---|---|---|---|---|---|---|
| 1 | positive | `Repetitive Nod + ''Okay''` | 0.06 (7) | 0.08 (10) | **0.74 (90)** | 0.12 (15) | vs. chance selection |
| 2 | positive | `Single Nod + ''Okay''` | 0.07 (8) | 0.16 (20) | **0.59 (72)** | 0.18 (22) | $\chi^2$ (df=3, N=366) |
| 3 | positive | `Single Nod + Eyebrow Raise` | 0.02 (2) | 0.04 (5) | **0.88 (107)** | 0.07 (8) | = 463.51, $p < 0.001$ |
| 4 | negative | `Eyebrow Raise + ''Was?''` | **0.51 (62)** | 0.09 (11) | 0.20 (24) | 0.20 (25) | vs. chance selection |
| 5 | negative | `Frown + Tilt` | **0.55 (67)** | 0.09 (11) | 0.28 (34) | 0.07 (9) | $\chi^2$ (df=3, N=365) |
| 6 | negative | `Frown + Tilt + ''Was?''` | **0.58 (71)** | 0.08 (10) | 0.21 (26) | 0.12 (15) | = 188.22, $p < 0.001$ |
| 7 | neutral | `IDLE` | 0.04 (5) | 0.07 (8) | 0.26 (32) | **0.63 (76)** | |

virtual Furhat improves the user experience. Additionally, they demonstrate that an overhearer online experiment setup is suitable to research the perception of Furhat's non-verbal behavior.

Heylen et al. [5] summarize studies that let 60 French adults assign functions to 21 backchannels or combinations of backchannels performed by the virtual human *Greta* in short video clips. They found that the function *understand* had been assigned most frequently to the expressions "nod", "nod and smile" and "nod and raise eyebrows". The *don't understand* function has mostly been referred to "frown", "tilt and frown" and "raise left eyebrow". "Tilt" on its own has rarely been classified as *don't understand.*

Inoue et al. [6] conducted an evaluation of their autonomous attentive listening system for the *ERICA* robot and compared it to a WOZ setting, i.e., the robot being controlled by a human. In the autonomous and WOZ condition, verbal backchannels, repeats, elaborating questions, assessments, generic sentimental responses, and generic responses were produced. 20 elderly Japanese interacted with *ERICA* in both conditions. The authors conclude that the autonomous system produces appropriate basic listening behavior, comparable to the WOZ condition, but can be improved in terms of showing empathy and other social skills.

Our study focuses on interactions in German and the distinction between positive and negative grounding acts, i.e., whether an overhearer can correctly identify the state of understanding that an embodied agent is trying to convey. To the best of our knowledge, our work is novel in its investigation of a virtual agent's listener feedback through facial expressions and head movements for the German language.

## 3 METHOD

### 3.1 Backchannel Design

To explore whether adults can recognize an embodied virtual agent's internal state by interpreting its backchannel behavior, we implement combinations of verbal and non-verbal backchannels for adult participants to categorize. The 6 backchannel combinations are summarized in Table 1. We use the virtual Furhat robot in preparation for future studies in which we want to incorporate such grounding displays.

The design of the tested feedback strategies builds on Clark and Brennan [2]'s theory of grounding. We consider one condition each for representing successful (*positive*) and unsuccessful (*negative*) grounding as well as a neutral baseline:

1) Positive: The agent signals understanding (acknowledgement act)
2) Negative: The agent signals non-understanding or confusion (request for repair act)
3) Neutral: The agent shows no particular reaction (baseline)

As baseline, Furhat stays in the IDLE state in which the embodied virtual agent performs only microexpressions like blinking and no other dedicated reaction to an utterance. The positive and negative conditions each involve one of three different combinations of backchannels to portray the intended grounding act. For signalling the positive and negative grounding states we consider combinations of head movements, facial expressions, and verbal expressions based on previous studies. Additionally, we are constrained by Furhat's features and limitations when choosing and implementing the backchannels.[2]

Clark and Brennan [2], Visser et al. [15] and Heylen et al. [5] mention nodding as an act of acknowledgement. Two versions of nodding are to be evaluated: single, the more commonly researched nod, and repetitive nodding, verified in Poggi et al. [11].

Raising both eyebrows had been perceived as a state of disbelieve, if accompanied by a nod as understanding, and raising just one eyebrow as not understanding [5]. In order to directly compare single and repetitive nodding, but also the presence of eyebrow raising vs. no eyebrow raising, the facial expression is only part of one positive setting design. In the confusion condition, due to technical limitations of the virtual agent, we chose the less appropriate raising of both eyebrows for setting 4. Heylen et al. [5] propose the frown alone or in combination with a head tilt as a non-understanding gesture. We select the version of the frown and head tilt combination for setting 5 and 6.

Allwood and Cerrato [1] have demonstrated that it is rare for humans to produce gestural feedback without accompanying vocal or verbal feedback. Therefore, four of the seven conditions incorporate verbal backchannels. Settings 1 and 2 that compare single and repetitive nodding, and settings 4 and 6 that compare two types of confusion expressions, are paired with a verbal backchannel. Clark and Brennan [2] mention "uh-huh", "m" and "yeah" and Visser et al. [15] mention "okay", "right" and "uh-huh" as examples of acknowledging backchannels. Because the Furhat voice in use only allows the generation of lexical backchannels, the decision fell on Visser et al. [15]'s proposed "okay" and Kopp et al. [8]'s mentioned "What?", translated to the German "Was?".

## 3.2 Interaction Domain and Backchannel Timing

The seven feedback strategies are applied to four spoken interactions in which a speaker tells a story and an embodied agent takes the listener role. Different parts of the recording were used to ensure that the content or quality of the audio does not influence the rating, since the semantic context of non-verbal feedback, among others, can highly influence its interpretation [5]. A 25 year-old male German native speaker was recorded talking about the Covid-19 pandemic and four parts of the audio recording were selected as stimuli. Humans process speech incrementally, with backchannels occurring during the speaker's turn [12], mostly timed after grammatical clauses instead of pauses [3]. We selected the four audio snippets in a way that they contained a backchannel relevant space [4] at which we then initialized the respective backchannel. The backchannel reactions have a duration of 2-5 seconds each and are initiated when a speaker pause is expected. The content of the audio recordings is specifically not task-oriented, i.e., the speaker does not directly ask for an acknowledgement or negative evidence from the listener and the content of the utterances is arbitrary. During recording, a listener was present to mimic a realistic story-telling setting for the speaker.

---

[2]For this study, we have used the Furhat SDK 1.26.0. Note that since the collection of this data, a new version (2.0.0) has been published that allows for more fine-grained implementation of some of the gestures and verbal utterances that we have used.

### 3.3 Video Item and Survey Design

In the overhearer study, participants only see the embodied agent listening to the speaker in an angled position, as shown in Figure 1.[3] The videos are created by playing and screen-recording the implemented skill that contains the backchannel combinations at pre-defined positions. Before and after playing the backchannel Furhat is displaying an IDLE state with only microexpressions. We pair each of the 4 audios with each of the 7 backchannel settings to create a total of 28 different video items that have a duration of 10-15 seconds each.

We follow Mirnig et al. [10] and Heylen et al. [5] in designing a closed-question survey in which participants choose one out of four options, displayed in Table 2.[4] Participants are asked to choose which description best fits the agent's internal state.

Instead of using labels like *confusion* and *understanding*, we phrase the options as full sentences that additionally contain a description of the kind of reaction that the agent seems to elicit. For example, a negative grounding feedback should ideally encourage the speaker to repeat or rephrase what they are saying; a request for repair act. The neutral condition, which is lacking any type of feedback reaction, does not portray an internal state and will not be included in any accuracy calculations. We expect the neutral setting to be interpreted as 'unattentive'. A fourth option to represent the 'cancel' grounding act [15] is added (prompt 4 'disinterest') and is expected to raise complexity of the closed-question format. We expect none of the settings to be interpreted as 'disinterest'. Participants could optionally comment on each video item via a free text field.

Table 2. Prompts that participants were given to rate the different backchannels. Prompts 1 and 2 encode the positive and negative grounding acts, respectively. Prompts were presented in random order for each new stimulus. The Furhat robot received the name "Bodo", which is used in the prompts.

| |
|---|
| 1) Bodo seems to understand, he wants the speaker to continue. |
| 2) Bodo seems confused, he needs another explanation. |
| 3) Bodo seems unattentive/absent. |
| 4) Bodo seems disinterested, he wants to end the conversation. |

In order to reduce the duration of the experiment for a single participant, the items were divided into two groups; each participant was randomly assigned to one of the groups. Each group saw each of the seven feedback settings twice. Group 1 heard audio 1 and 3 once more than audio 2 and 4. Group 2 heard audio 2 and 4 once more than audio 1 and 3.

The data was collected as part of a Bachelor thesis and part of the participants were a convenience sample of people wishing to contribute to the student's research. Further participants were recruited via a university portal for scientific experiments and reimbursed via course credit. We used the software PsyToolkit [13, 14] to set up the online survey. All our materials can be found in our documentation.

## 4 RESULTS

64 participants filled out the survey. We excluded one submission because it was incomplete and one submission that was completed in 5 minutes, which we deemed too little time to watch the videos and answer all the questions.[5] We analyze the remaining 62 submissions (45 female, 20 male, 1 non-binary; age: M=26.34, SD=8.87, min=19, max=59). All

---

[3]Further details of the settings, such as speech rate and gesture strength, can be found in the implementation that we release with our documentation: https://anonymous.4open.science/r/CanYouTellThatImConfused

[4]The German original options can be found in our documentation.

[5]The average completion time of the remaining participants was about 22 minutes, with a minimum of 11 minutes.

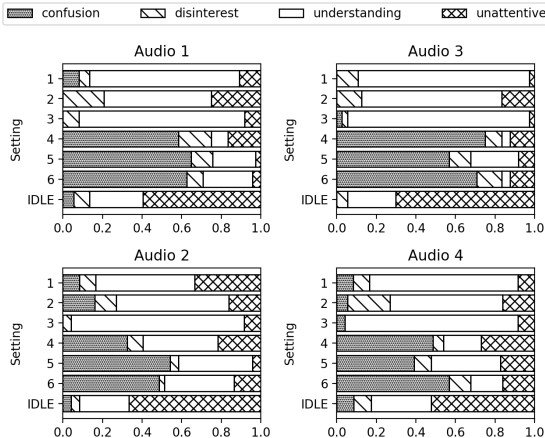

Fig. 2. Distributions of the answers for all backchannel settings. For descriptions of the settings see Table 1

participants reported to be native German speakers, including four bilingual speakers (two Turkish, one Hungarian and one Greek). 37 participants had been assigned to group 1 and 24 to group 2.

Table 1 summarizes all participants' ratings. A general pattern can be distinguished in which the neutral condition (7) has been identified as mostly *unattentive*, the negative conditions (4-6) as *confusion* and the positive conditions (1-3) as *understanding*. None of the settings display a large proportion of interpretations as *disinterest*. All settings generally have the highest count for their intended grounding act and better than chance-level (0.25), though the accuracy across settings varies. Setting 3, which is the condition of a *single nod* and *eyebrow raise*, has the highest accuracy (0.88).

The three conditions signaling understanding have overall been recognized with a higher accuracy than the three conditions displaying confusion (0.73 vs. 0.55). The backchannels communicating understanding appear to be less ambiguous than the backchannels generated for confusion. The repetitive nod with verbal backchannel (VB) has a higher accuracy than the single nod with VB. Consequently, the repetitive nod, as proposed in [11], appears to be less ambigious than the single nod.

In case the feedback intended to communicate understanding was misinterpreted, it was mostly classified as *unattentive* or *disinterest*, but very rarely as *confusion*. The acts signaling understanding were hence sometimes perceived as failed grounding, but rather due to an impression of absence of attention. In contrast, the feedback displaying confusion was mainly misinterpreted as understanding, i.e., successful grounding.

Figure 2 visualizes the distributions of answers for each audio condition separately. To gain further insight into whether the audio samples affect the recognition rate of the backchannels, the accuracy for the understanding and confusion condition are calculated for each audio (cf. Table 3). Overall, the understanding condition has been recognized best. The four different audio snippets do differ with respect to recognition accuracy, with audio 3 having the best results. We discuss this difference further in Section 5.

## 5 DISCUSSION

We observe that participants tend to interpret feedback rather as positive than negative, meaning that the backchannels have rarely been interpreted as displaying confusion when it was not intended. The conditions of positive feedback have generally been recognized more often than the condition of negative or neutral feedback. It is an open question

Table 3. Recognition accuracy of intended meanings across audios

| audio | understanding | confusion | total |
|---|---|---|---|
| 1 | 0.73 | 0.62 | 0.68 |
| 2 | 0.64 | 0.44 | 0.53 |
| 3 | **0.85** | **0.66** | **0.76** |
| 4 | 0.71 | 0.49 | 0.59 |

whether these differences derive from the quality of our backchannel implementation or whether specific backchannels are less ambiguous than others. Furthermore, seeing that one audio condition has had a much higher recognition rate than the others, the particular content, intonation, or other speech properties may have played an additional role.

The IDLE condition in which Furhat does not react to the speaker's utterance has been judged as *understanding* in one fourth of the cases (0.26, in 0.63 as unattentive). On the one hand, this supports the assumption that the absence of negative feedback can be interpreted as successful grounding [3]. On the other hand these results demonstrate the need of explicit positive evidence for successful grounding in HRI, as it is the case in human conversation [2], since the baseline condition has predominantly not been interpreted as continuing attention (successful grounding) but rather as mental absence (unsuccessful grounding). Similarly, the *confusion* backchannels were often misrecognized as signalling understanding (0.20-0.28, 0.51-0.58 as confusion). The frequent misinterpretations can be attributed to Clark and Brennan [2]'s most common type of acknowledgement acts "assessments", i.e., an acknowledgment that values the presented information. The VB "Was?" might have been perceived as such, expressing surprise or other emotions. Although we received only a few comments by our participants, some of their feedback supports this assumption. In terms of individual settings, the positive backchannel consisting of a single nod with an eyebrow raise (setting 3), was identified correctly by most participants (0.88). For audio 3, the accuracy comes up to 0.92.

## 6 CONCLUSION

We have presented an overhearer study investigating whether successful and unsuccessful grounding backchannels can be recognized in German interactions with the Furhat virtual agent. Our results show that all three categories can be recognized in the majority of the cases, which is consistent with our hypothesis. At the same time, there is much room for improvement and free-text comments by the participants hint to issues in timing and contextual mismatches. Participants in our study assumed *understanding* in many cases in which it was not intended, but rarely misinterpreted backchannels as *not understanding*, if not intended. Moreover our results demonstrate that a variety of different behaviors can be used with the Furhat agent to signal positive understanding in an efficient undisruptive way.

Like Zheng and Meng [17] and Mirnig et al. [10], we conclude that it is worth conducting tests in advance to verify the quality of the design and implementation of the chosen listener responses. Additionally, the appropriateness of newly created audio stimuli and backchannel timings should be verified either through pre-experiments or a comparable condition, e.g., a WOZ setting (as done by Inoue et al. [6]). Furthermore, it is worth conducting more research that aims to create guidelines for a straightforward implementation of smooth and natural custom gestures for Furhat, comparing the physical and virtual versions.

Open questions for future work include investigating whether the same backchannels that can be recognized by an uninvolved listener can also serve a direct listener in a task-oriented scenario, and experimenting with more fine-grained gesture and voice settings that a newer version of the embodied agent provides.

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
