# OpenReview forum: "Can you tell that I'm confused? An overhearer study for German backchannels by an embodied agent"
_ACM.org/ICMI/2022/Workshop/GENEA — GENEA Challenge & Workshop 2022 Workshopproceeding_

### Official Review · Reviewer_J3n3 · 2022-08-06
**Interesting paper that presents a work that is a good contribution to the community but still needs some design thinking especially regarding evaluation**

**Rating:** 7
**Confidence:** 4

**Review:**

The authors present a study on backchannel behaviours using a FurHat robot where they verified the accuracy of a given set of backchannel behaviours for the understanding (positive) and confusion (negative) groundings. The backchannel behaviours were designed based on literature and on FurHat's own limitations. The study shows that the behaviours were mostly correctly understood, with the positive ones working slightly better than the negative ones.

The paper is well written and easy to understand.
I enjoyed reading the introductory section and literature review.
However I feel a little skeptical about the soundness of the study for these particular behaviours.
The authors do mention they plan to perform a new study where the participants are direct listeners, and not uninvolved ones.
The main question I have is whether the fact that the participants are able to think about the agent's behaviour, and assess it post-interaction, does also mean that they would understand it properly and immediately during the actual interaction.
I understand that is one of the major challenges in such a study, so I must add that my doubts are not towards the authors or their capabilities, but regarding the overall study goal and study design.
From my previous thoughts and experiments on backchannel behaviour, it seems like a more direct observation methodology may provide a higher fidelity of results - i.e., if the backchannels are correctly interpreted during interaction, then it will be fluid, otherwise, the interaction may seem broken or confusing to the user, therefore leading to unexpected pauses or breaks.
In that sense it may seem like a good idea to evaluate indirectly - a set of participants A interact with the robot directly, the robot performing backchannel behaviours.
The actually study is then having another set of participants B who watch the recordings of the interactions between robot and A, and evaluate whether or not the participants A understood and reacted promtly to the robot's backchanneling behaviours.

My recommendation is for acceptance of this paper into the workshop but hoping that the authors may comment on these issues during the presentation of their work.

---

### Official Review · Reviewer_37K6 · 2022-08-15
**description of a study on backchannel perception. The backchannels are displayed on the Furhat robot.**

**Rating:** 6
**Confidence:** 5

**Review:**

The paper presents a perceptive study of the Furhat robot displaying backchannels while a 'voice' is telling a story.
The authors aim to validate the perception of bakchannel meaning.
The choice of the nonverbal behaviors is grounded on previous studies.
The state of the art section is ok. But it is missing some references. The generation of backchannels has been well studied for ECAs and robots. I recommend the authors to look at the works of Gratch (Virtual Rapport 2.0) and of Buschmeier and Kopp, to name a few.
I would also suggest to distinguish between works studying the perception of backchannel signals (eg [5]) and works aiming to generate where to place a backchannel (eg [7]).
It is not clear to me why the authors added the function 'disinterest'. What was there hypothesis for this function? Why not considering the function 'interest' to balance the choice of functions?
It is not clear to me how the backchannels have been placed in the stimuli. Backchannels do not appear systematically on pauses.
How many backchannels were produced in each stimulus. It seems there is only one backchannel. Thus each stimulus must be extremely short. It would be worth to indicate their length.
The authors should mitigate their conclusion: the perception of backchannels may vary with more information on the context of the interaction (cf work on rapport building).

---

### Decision · Program_Chairs · 2022-08-20

**Decision:**

Accept (Workshop proceeding)

**Comment:**

The state of the art section is okay, but missing some references. The paper is well written and easy to understand. As indicated by one reviewer, it is unclear why the function ‘disinterest’ was added. It is also not clear to this reviewer how the backchannels have been placed in the stimuli. Another problem is that the length of the stimuli is not reported. The second reviewer calls for another study design, and wishes for the authors to respond to their comments during the presentation of their work.